# Letter to Matter and Various Incomprehensibilities—The Effective Ethicality of Scientific and Humanistic Interdisciplinarity [†]

**Gianfranco Minati**

Italian Systems Society, 20161 Milan, Italy; gianfranco.minati@AIRS.it; Tel.: +39-02-6620-2417
† In memory of Eliano Pessa.

**Abstract:** The article is based on the dual concepts of theoretical incompleteness in systems science and theoretical incomprehensibility in philosophy previously introduced in the literature. Issues of incompleteness relate to the logical openness of complexity models in their nonequivalence and necessary non-zippable incompletable multiplicity. This concerns the quasi-ness of phenomena and the constructivist nature of models. Theoretically, incomprehensibility is considered in different ways, such as the inexhaustible multiplicity of the constructivist reality corresponding to the logical openness of both the world and of understanding itself and in reference to incomprehensibilities such as questions like the divide between species, cognitive systems, and after-death experience (if any). In conjunction with the need for non-classic, interdisciplinary approaches in science to deal with complexity, unanswerable questions need suitable scientifically updated philosophical reasoning in interdisciplinary humanistic backgrounds to allow for new social representations, understandings, and plausible social imaginary. Such approaches more properly allow for effective philosophical representations of the world. Knowing how to deal with these issues through philosophical reasoning updated to the current scientific levels and humanistic interdisciplinarity allows for higher levels of awareness and new comprehensive philosophical understanding, introducing new powerful social imaginary. Such new philosophical reasoning is expected to allow a conception of the world that is no longer just utilitarian, but theoretically necessarily, and not only concessively respectful of diversity with significant, even self-ethical effects.

**Keywords:** alienation; cognitive; impassability; incomprehensibility; interdisciplinarity; matter; understanding

## 1. Introduction

In a previous contribution dedicated to theoretical incomprehensibility [1], we considered its conceptual duality alongside theoretical incompleteness [2–4].

We considered theoretical incomprehensibility in different ways. One relates to the inexhaustible multiplicity of the constructivist reality corresponding to the *logical openness* [5], [6] (pp. 47–51, 111–112) of both the world and of the understanding itself. We mention here how the concept of logical openness is clarified by the contrast with logical closedness. A model is logically closed when the relationships between its state variables are complete and a complete formal description is available; the interactions between the system and its environment are described in a complete and explicit and available way; and knowledge of previous conditions allows us to deduce all the possible states that the system can assume. The model is then completely computable. The concept of logical openness refers to cases where there is a violation of at least one of the three previous conditions. Theoretical incompleteness relates to cases when logical openness is not temporary, i.e., due to a momentary lack of suitable tools allowing logical closedness, but theoretical since the phenomena presents continuous structural changes as for complex systems requiring multiple nonequivalent modeling [6] (pp. 64–85), [7] (pp. 201–204), in multi- and interdisciplinary ways. This is the situation when the system to be studied is

so complex, e.g., constituted by multiple processes of emergence, that it is impossible, in principle, to fully describe it using either a single model or a fixed sequence of models.

Logically closed models in any quantity and configuration cannot completely represent the phenomenon, while logical openness represents the dynamics of significant properties such as combinations of different coherences. This particularly applies to aspects of incompleteness and the quasi-ness of phenomena of emergence that cannot be modeled or zipped in a single model [8]. In the usual understandings, based on Newtonian non-complexity, comprehensibility almost coincides with approaches and models as complete knowledge characterized by logical closedness.

Dealing with complexity requires approaches based on logical openness. This is a matter of having cognitive strategies and approaches suitable to deal with multiple equivalences and quasi-ness of multiple emergences having instabilities of properties; non-binarities; non-invariance; nonregular alternations of degeneration and recovery establishing systems that are not always systems, not only systems, and not always the same systems; but, on the other hand, are suitable to represent, deal, cohabit, combine with, and use theoretical incomprehensibility. The latter is the case of imaginary numbers and the unreality of real numbers [9] (for instance, we can use the term $\sqrt{2}$ and its symbolic properties, but not the explicit result of the computation).

We considered how the issue of theoretical incomprehensibility is related to the impossibility of modeling in completed, zipped, deducible, symbolically computable ways. Furthermore, the issue of theoretical incomprehensibility cannot be considered without reference to an agent endowed with cognitive abilities. In the same contribution [1], we considered the issue of incomprehensibility as related to existential questions such as: What is the purpose of life? Is there anything after death? We elaborated on such questions considered as incommensurable with rational answers. We also mentioned the usage of the illative sense [10,11], theoretical reasoning that judges the validity of inferences.

In this contribution, we further elaborate on issues related to incomprehensibility, such as those related and compatible or even partially specified by various specific incomprehensibilities, such as considering the cohabitant alienation among different living species. Other cases are given by the impassability between different cognitive systems (we can know everything about bats, but we will never be a bat [12]), with the passability, however, being partial even between cognitive systems of the same species, and by existential questions.

Furthermore, the emergent identities always differentiate between living beings endorsed with the same kind of cognitive system. We consider the case of approaches to levels of *comprehension of the incomprehensibility* when its possible representation requires describing ourselves as alien from our human species. This was the case for the first autopsies, and now for the design of simulated human-like behavior in robotics and artificial intelligence (AI), requiring external, uninvolved points of view, without taking anything for granted, as for the still neglected creative role of the unconscious in AI. In the 1980s, Marvin Lee Minsky related the concept of the unconscious to memory when he wrote: "Usually, we have no conscious sense of this happening, and we never use words like 'memory' or 'remembering' when the process works quickly and quietly; instead, we speak of 'seeing' or 'recognizing' or 'knowing'. This is because such processes leave too few traces for the rest of the mind to contemplate; accordingly, such processes are unconscious, because consciousness requires short-term memory. It is only when a recognition involves substantial time and effort that we speak of 'remembering'." [13] (p. 154).

Such approaches without, or with reduced, self-involvement allow us to further elaborate on issues of cognitive impassability in general.

Such approaches allow us to make existential issues less incommensurable, more philosophically representable through comparisons and possible levels of compatibilities, and acceptable as related to emergences of life and identity from phenomena of impassability, localities, and constraints. *The availability of such philosophical understandings is expected to have positive effects in terms of allowing new, less egocentric conceptions of the world, open to*

*conceiving incomprehensibilities and not just tolerating them, for a new relationship with living beings and what is alien to us*. However, being subjective doesn't imply being egoistic or egocentric: feeling to be myself is not negotiable.

Differently from the issues presented in [2], where the incompleteness can be more properly represented, dealing with the theoretical incomprehensibility of existential questions requires the use of cultural, humanistic resources such as arts, literature, poetry, religion, and philosophical reasoning. This article considers the last approach, and the purpose is to outline related philosophical reasoning based on contemporary scientific conceptions, different from the still used classical ones, offering new conceptual visions of matter and allowing new forms of understandability of incomprehensibilities and impassability such as those related to existential questions. As for complexity, a single discipline or a single approach is not enough [6,7]. In the same way, the living cannot be considered only as a matter of biology, chemistry, or physics.

Trans-disciplinarity may be intended to occur when considering acquired systemic properties, e.g., coherence, self-organization, shapes, and behaviors in general, regardless of any disciplinary context, such as when dealing with complex systems. Multi-disciplinarity relates to the simultaneous usage of different disciplinary representations of the same phenomenon, dealt with distinct from each other disciplinary, specialized, and well-distinguishable, however coordinated, approaches. It is a matter of separated competences and specializations.

Scientific interdisciplinarity has long been considered in systems science. Interdisciplinarity is intended to take place, for instance, when problems and solutions from one discipline are used in another; when the same models and approaches are used in different disciplines through a change in the meaning of variables; when a problem is converted into another one, e.g., from military to economic or from geometrical to algebraic; and when setting biologically inspired strategies in economics or defense or when teaching a discipline while talking of another, e.g., teaching history when talking of geography or archeology. Trans-disciplinarity is disciplinarily acontextual, multi-disciplinarity is supposed to just use and not elaborate disciplinary contributions, while interdisciplinarity allows one to elaborate, share, and test disciplinary approaches.

We think that a similar interdisciplinarity may be considered in nonscientific domains, particularly among the arts, culture, literature, poetry, and philosophy; see, for instance, [14]. This is also the case for interreligious dialogue[1] as a particular case of humanistic interdisciplinary practice [15]. Such interdisciplinarity may be intended to be constituted of different representations of the same meanings, human and existential problems, hopes, and sentiments. Philosophical reasoning may be intended as a suitable meta-level not characterized by its own object, but by its own level and combination of scientific and humanistic interdisciplinarity, as emerged in a conversation with the philosopher Lucia Urbani Ulivi, who I would like to thank.

Examples of humanistic interdisciplinarity include film versions of novels or their soundtracks; songs and lyrics in music (note how the lyrics have become predominant over the music, while in the Baroque era, for example, words were mostly of interest for their musicality and the singers were considered "musical instruments"); multimedia advertising; and stereotypical use of slogans from songs, films, and advertising phrases. Another example is the predominance of functionality (for maintenance and safety) in design and in architecture, leading to a reduction of details (in contrast, for example, to the Baroque era with its predominance of details). Furthermore, baroque architecture and baroque music share the same attention to detail. Humanistic interdisciplinarity, with its multiple and mutual representations, makes conceptually available philosophical

---

[1]　See, for instance, *The* Pontifical *Council for Interreligious Dialogue* https://www.vatican.va/roman_curia/pontifical_councils/interelg/documents/rc_pc_interelg_pro_20051996_en.html (accessed on 4 March 2021).

issues such as beauty, death, fear and god accessible[2], as well as importance, joy, life, nobility, purity, strength, and the universe. Some of these issues have been studied, at different levels, in scientifically interdisciplinary ways (for example, life in biology, matter in chemistry, the universe in physics, and understanding in cognitive sciences), but often the humanistic and philosophical aspects have remained incoherent and unchanged.

Scientific and humanistic interdisciplinarity, reciprocally using, sharing, and negotiating concepts and meanings, constitute philosophical reasoning, sharing approaches and understandings such as within the mesoscopic description level of description, areas of continuous negotiation between the micro and the macro [8]. Examples include philosophical reasoning, usually still based on a classical Newtonian understanding of the world with the assumption of general validity of anticipation, comprehensibility, completeness, computability, computable probability, decidability, error-freeness[3], formalization[4], gravitational forces as an attraction between masses (while in Einstein's general relativity, gravitation is an attribute of space-time: masses deform space-time and bodies move in trajectories determined by the geometry of space-time), non-contradictoriness (while in physics, there are the so-called uncertainty principles, wave-particle duality, and quasiparticles that share traditional particle properties, with the exception of localization), optimization, precision, pre-relativistic time, regulation, reversibility, separability, solvability, and stationariness [18].

This interdisciplinarity is seen, for example, in the fact that, on the scientific side, approaches, methodologies, and applications are chosen to be coherent with the cultural and philosophical modalities and reasoning in use, and, on the other hand, social imaginary and social views are developed as cultural concepts that elaborate and use technologies and conceivable possibilities. We figure out a philosophy of complexity dealing with concepts such as coherence, dynamics of properties, emergence, emerging computation, equivalences, from fields to validity regimes, singularities and uniqueness, structural multiplicity, network properties, non-separability, non-computable probability, non-decidability and non-prescribability, non-invasiveness, quasi-ness, remote synchronization, stigmergy, structural dynamics, use and not only respect for the degrees of freedom, and usage of weak forces, all translated into social culture.

Separating, i.e., ignoring the double nature of this interdisciplinarity (scientific and humanistic) is a fact of the new reductionism and will have serious social consequences, up to and including technocracies, manipulation, and self-referential logic. The humanistic and scientific interdisciplinarity overlaps with human anthropology. In the case of separation, there is the establishment of internally inconsistent and unaligned social systems, as has been and is the case with technologically very advanced societies (e.g., issues of human rights and the relationship with nature). We mention how such separation may be considered as represented by the contrast between the first and second Wittgenstein

---

2    In Genesis (see Table 2), it is written: *Let us make man in our image, after our likeness*. The Christian god, considered representable, contrary to the original Jewish culture, is portrayed as male and elderly, as a sign of wisdom and as in the Sistine chapel: rather, he is represented in *our* image and likeness.

3    A logical or material device intended as completely and identically iterable, repeatable based on logical closeness, is reliable but not considerable as intelligent just because it never make mistakes, i.e., deviations from the expected procedural behavior. It is a matter of first-order cybernetic *trivial machines* whose operations are not influenced by the outcomes of previous operations performed by the machine itself. A device that does not perfectly iterate but allows events of logical openness, seen as imprecisions and mistakes in its autonomy as an *inaccurate machine*— context-dependent, making nonequivalent abductions, able to learn, repair, and reformulate itself—can be considered to have significant levels of intelligence [16].

4    In this regard, we can remember that Feynman considered the 'Greek' versus 'Babylonian' mathematics. The 'Greek' approach to mathematics is characterized by the tendency to arrange theories on an axiomatic basis, whereas Feynman writes: "What I have called the Babylonian idea is to say, 'I happen to know this, and I happen to know that, and maybe I know that; and I work everything out from there. Tomorrow I may forget that this is true, but remember that something else is true, so I can reconstruct it all again. I am never quite sure of where I am supposed to begin or where I am supposed to end. I just remember enough all the time so that as the memory fades and some of the pieces fall out I can put the thing back together again every day'." [17] (p. 45). He also writes, "In physics we need the Babylonian method, and not the Euclidian or Greek method" [17] (p. 47). The end of the so-called *Bourbaki program* (1935–1998), aiming at a completely formalized self-contained treatment of the core areas of modern mathematics based on set theory, introduced a new understanding of formalization [7] (pp. 6, 192).

related to the turn from formal logic to ordinary language, reflections on mathematics and psychology, and skepticism of philosophy's pretensions.

In this conceptual framework, we may consider extensions of incompleteness and quasi-ness in humanistic and philosophical contexts—for instance, when changes of predominance occur in representation and significance. Other cases relate to equivalences and non-equivalences, redundancies and multiplicity.

The problems of complexity require the interdisciplinary usage of multiple approaches and models[5]; in the same way, themes related to the philosophical understanding of the mind-body relationship; cognitive abilities; impassability between coexisting species, between versions of the same cognitive system and related emergent identities, and between life and death, require multiple humanistic interrelated views.

The general hypothesis of the article is that considering incomprehensibilities requires scientific and humanistic interdisciplinary philosophical reasoning, allowing for an integrated, ethical, and comprehensive understanding of the world. We consider how the considering of alienation and impassability could facilitate the social adoption of such reasoning and suitable social imaginary.

Therefore, Section 2 uses the theme and the epistolary style of a letter to someone, having numerous precedents in literature, music, and cinematography[6]. The section presents a cultural framework based on the constructivism of the logical openness with some exemplificative religious mentions, devoted to recognizing the central role of matter in a nonreductive or materialist way. The issues mentioned above are then elaborated on within the metaphorical framework of a hypothetical letter to matter[7], based on an understanding that living beings acquire emergent properties as materializations of constraints, properties, and the evolutionary requirements of nonliving and living (metabolic/repair) matter. The common thread between matter and cognitive abilities is considered, since matter is regarded as understanding itself [1], [7] (pp. 257–258).

Section 3 uses the philosophical reasoning developed in systems science to elaborate on the themes mentioned above, such as the cohabitant alienation between living beings, the impassability between different cognitive systems, irreversibility in general, and the impassability and irreversibility between life and death. Knowing how to deal with these issues through philosophical reasoning updated to the current scientific levels and humanistic interdisciplinarity allows for higher levels of awareness and new comprehensive philosophical understanding, with a profound overall cultural meaning introducing a new and powerful social imaginary. Such a new philosophical understanding is expected to allow comprehensive conceptions of the world that are no longer just utilitarian, but theoretically necessary and not concessively respectful of diversity with significant, even self-ethical effects.

We conclude by stressing the need to introduce new, updated philosophical concepts, allowing for philosophical reasoning suitable to better realize the current scientific understanding and cultural incomprehensibilities, and introduce a new, powerful social imaginary. Such updated interdisciplinary philosophical reasoning should be not be scientistic, but should integrate humanistic interdisciplinarity, as cultural, theoretical necessity and not as gracious concession. Such integration should not be considered final, but a generative process in itself. Table 1 summarizes the concepts and purposes of the article.

---

5 Through, for instance, the Dynamic Usage of Models (DYSAM) [6] (pp. 64–85), [7] (pp. 201–213), based on established approaches in the literature, such as *Ensemble learning* and *Evolutionary Game Theory* and considering the continuous explorations of available approaches.

6 See, for instance, the romance *Letters to Camondo* by Edmund de Waal, published in 2021 by Chatto & Windus; *Letter to You*, Bruce Springsteen's twentieth studio album, released in 2020 by Columbia Records; and *Letters to God*, a film directed and produced by David Nixon via Possibility Pictures.

7 From now on the word Matter will be capitalized when considered it as a metaphorical interlocutor. When appropriate in the discourse, we will reveal ourselves to it with the feminine *she/her* in the optional cultural context of assuming her as the mother of everything. The word Nature is a hasty assumption, but sufficient for the discursive level used here, as a concept having many dimensions and historically treated in philosophy by various authors indicating what exists. From a systemic understanding, matter may be intended as only *possessing* properties and their combinations, while Nature may be intended as matter with its *emergent acquired* properties, such as the condensed and its collective effects, and living evolving matter. *Matter is intended as a necessary level for Nature.*

**Table 1.** The central issues elaborated in the article.

| |
|---|
| Existential questions, having the nature of theoretical incomprehensibility, are comparable with issues of impassability between species, different cognitive systems, irreversibility and after death (how detectable, if any?). |
| Dealing with such issues of theoretical incomprehensibility requires scientific and humanistic interdisciplinarity leading to suitable scientifically updated philosophical reasonings and humanistic imaginary. |
| Such updated philosophical reasonings and imaginary with higher levels of awareness, allow theoretical (and non-only concessive) ethical, non-homo-centered comprehensive views of the world that are no longer just utilitarian, but respectful of diversity with a great ethical effect reflective of itself. |
| We realize that the significance and power of questions having theoretical incomprehensibility, such as existential and related to impassability between alienities and different cognitive systems, lie in considering, elaborating rather than ignoring them as unanswerable; in considering their logical openness, their theoretical insolvability to be continuously managed by suitable multiple (see note 5) philosophical reasoning in the framework of theoretical incompleteness, scientific and humanistic interrelated interdisciplinarity, rather than in improbable answers. |
| However, the thesis of this article is that the ethical, comprehensive understanding of the world is related, embedded, theoretically integrated in scientifically updated and humanistic interdisciplinary philosophical reasoning. We consider how the considering of alienations and impassability could facilitate the social adoption of such reasonings. |

We consider how the ethical, comprehensive understanding of the world is related, embedded, theoretically integrated in scientifically updated and humanistic interdisciplinary philosophical reasoning. We consider how incomprehensibilities such as alienation and impassability could facilitate the social adoption of such reasoning.

## 2. Letter to Matter. Understanding as the Duty of My Species

I know you want me that way . . .

I am writing to you without a good understanding of what exactly you are. However, the story will probably become clear during the telling.

At the beginning, and simplistically, we have figured you out as materiality, provided with mass, volume, and localization. You also have the qualities of biotic matter. By using cognitive resources that we have received from you and subsequently improved, we have considered different cognitive powers and the effectiveness of defining, hypothesizing, considering, and modeling you. We also have the cognitive need to state your existence [19], and to accept you.

There are now various levels of understanding of matter such as condensed matter in physics [20,21]. In a quantum understanding [22–25], you have come after the emptiness (quantum vacuum) assumed to be existing before you and having properties. We even think of you as invisible, i.e., dark matter, in the universe, which is assumed to be simplistic and closed, whereas it should be understood as open.

Perhaps you are a property of something else, inhabiting moments and thresholds, quasi-particles and quasi-matter. The search to define you is probably a poor strategy.

I am writing to you because is at these undefinable thresholds that you build up worlds. I am living in one of those worlds. I will call this world Nature (see footnote 7). In this world, you have assumed properties that we may name vital, so we may then speak of living matter. In this world, pieces of living matter must behave to keep their properties, i.e., they must eat, drink, reproduce, sleep, delay as much as possible thermodynamic death, etc. They must dissipate [26,27] and keep emergent properties, as initially conceived by Conwy Lloyd Morgan in 1923, and later described by the philosophers Charlie Dunbar Broad in 1925 and Arthur Oncken Lovejoy in 1927 [28–30].

In order to do that, this world that we name Nature, eventually in connection with other worlds and looking for its un-understandable purposes, has provided for its purposes living matter with levels of cognitive abilities.

I am writing to you because we have started to improperly use such cognitive resources in order to understand themselves, you, and ourselves. Living matter has begun to behave, enjoy, use, and refuse according to your purposes, some evolutionary and others incomprehensible to us, probably because you do not need us to comprehend them.

We assume that we understand what you need from us, but we have gone beyond. Is the ability to understand you embedded into you, and do we materialize or impersonate it? Now we hope and get joy and pleasure not only from you, but also from entities we have created such as music, paintings, and poems; is this something unexpected for you?

We owe you everything.

Since ancient times, thinkers like Epicurus, Giordano Bruno, Lucretius[8], and Tommaso Campanella have focused on your properties. I suppose that your properties represent a way by which becoming happens. Not *why*, but rather *how*. Everything must happen somehow. Neuronal changes do not constitute thought, but the way from which it emerges.

We build a life of differences from your purposes by slipping between, and by using what you want us to do.

Why do you need life? To understand yourself? Do you have to speed up some processes?

Do you want us to make non-natural changes?

You have made us to rejoice in what you need, but we have learned to rejoice in something else. Is this a deviation or do you want us to build new worlds?

Our lives are now combinations of delights in what you need and joy in what we have "autonomously" generated. As a price, we have begun to joy, hope, miss, and suffer: all this while you are indifferent, a loving mother to species and an insensitive stepmother to individuals. Death is now something other than the end of an assignment that we got from you. You need death.

We are realizing that this is the price of autonomy, but probably it is always you with another visage, another world.

We explore the borders of incomprehensibility as properties of something else.

By using areas of cognitive activity differently from natural prescriptions having well-defined purposes, we have become aware of our behavioral limitations.

It is possible to hypothesize an ideal process, a biological-cognitive converter, through which you make living matter recognize and desire, e.g., actions, behaviors, sensations, and shapes, such as through the effects of neurotransmitters such as dopamine. Well-known cases concern sexual attraction at reproductive age; attraction or repulsion to possible foods depending on their color, shape, and smell; judging behaviors as attractive or dangerous; and even the smile of a child.

We don't fully know how this conversion occurs; surely it is not mediated only by biochemical agents: perhaps you do not need our understanding.

Some belonging to our species, such as hermits, monks, and mystics, refuse the game and pursue a mystical path to play another game within yours or even to refuse life itself. Would the greatest freedom be to deny yourself and reject your game? Are you using a fact of autonomy or another fact of unconscious obedience?

Now we use the cognitive processes that you have provided us with in order to understand themselves with cognitive sciences.

However, what is the limit of our cognitive system?

---

8 See, for instance, the concept of *clinamen* (swervement, inclination, deviation), considered by Lucretius in *De rerum natura*, where in book II he deals with the qualities of atoms: in continuous, very fast motion in an unobstructed vacuum. The atoms move from top to bottom and, thanks to the clinamen, are defined with respect to the vertical: they bounce, meet, and aggregate; the diversity of their forms and the multiplicity of combinations generate the variety of things: "id facit exiguum *clinamen* principiorum nec regione loci certa nec tempore certo" (From that slight swervement of the elements in no fixed line of space, in no fixed time) [31].

Who am I? The being who desires what you need, disguises what you do not want, or understands your game?

I understand that you made me so as to eventually realize that your game had other purposes.

I cannot avoid playing your game, while understanding it at least a bit.

Probably, as for the emerging computation [32], we have to play the entire game before we can "know" it.

Should I wary of pleasure and even joy, as not mine? In the same way, I respond to the smile of a child and the kiss of a lover. I must love them, and you must not make fun of me; you are insensitive and ready to take them away from me at any time. I will try not to feel offended. Are you using me or am I using you? Or both? I will keep smiling as an act of my species; I will impersonate you over time. I now understand that I will belong to you for my entire life, even if at different levels, because what exists is from you even if not just from you.

However, I have acquired the hope to know and to understand you. Maybe I will know when I will be no more only a property of living matter.

I recognize you in the joy of a mother, in the rattling breath of a dying person, in the rising of the sun, in the dialogue between the moon and the mountains, in the look of agony, and in remembering joy.

In a pantheistic vision, there is nothing else but you, beyond all the acquired properties. However, I will be worthy of you, as the duty of my species.

## 3. Various Incomprehensibilities

Drunk on moments, we continue to extract them from our amphora of life as if it has no limit, and asking where it came from makes no sense.

Phrases said want to be retold; moments past and present come to meet us and hope to be lived and relived.

The issue we are dealing with here is adjacent to that of theoretical incomprehensibility [1]. The theme we have in mind relates to generic processes of becoming other, mutually immeasurable, incomprehensible as an acquisition of nonlinearity, adaptations, evolution, irreversibility and irreducibility, mutations, restructuring, self-organization and emergence, structural dynamics, structural transformations, and transitions (for example, of phase).

One aspect of becoming other is that the observer for whom something becomes other must conceive both the pre-other and the other that it has become. This paradoxically assumes the necessity of assuming the comprehensibility of the manifestation of incomprehensibility.

Let us consider, in an introductory way, cases in which there may be coexistence between the observer and other presumed absolutely alien observers, irreducible to each other, as in the case of life forms. A particular case occurs above all when the constitution of the other is presupposed as a substitute of the pre-other, a substitution that is, however, inconceivable, i.e., becoming something else yet conceivable. In the latter case, becoming other and the other itself are mutually inconceivable, only conceivable in an abstract way, for various reasons such as the assumption of forms of continuity and replicability, cultural, existential, and religious. The latter two cases concern, for instance, life after death (or, rather, reaching a certain level of the dying process [33]).

There are infinite cultural, philosophical, and religious approaches in this regard, up to the belief that the other (for example, the soul or the spirit) is dual to the living (for example, generated by the living), or that it is even previous, or, again, affirming itself with the end of biological life (e.g., with the cessation of identity generation, as considered below).

Identities may be intended in physical worlds as non-equivalences, specificities, and singularities whose temporariness results in emergent phenomena.

A crucial issue concerns existence and its conceivability as a condition for having a role, including that of not being there [19,34]. In Table 2, I mention inspirations coming from my

own culture. However, it is a social vision of the world and life is still active and supporting understanding. Surely this view has several different corresponding approaches in other cultures and religious understandings. As for disciplines, the interdisciplinarity should not be reduced to coexistence and tolerance only, but should be active, mutual, interactive and suitable to make emergent higher levels useful for human beings and our whole lived context.

**Table 2.** Inspiration from the Hebrew Torah.

| From the Torah, Bereshit: |
| --- |
| [1, 26] *And the Lord said "Let us make man in our image, after our likeness, and they shall rule over the fish of the sea and over the fowl of the heaven and over the animals and over all the earth and over all the creeping things that creep upon the earth".* <br> [2, 19] *And the Lord formed from the earth every beast of the field and every fowl of the heavens, and He brought [it] to man to see what he would call it, and whatever the man called each living thing, that was its name* (the Lord is assumed to not know in advance the names the man will give to them). <br> The Lord created the human having forms of independence from him. Why? Do we have a mission, special among the living beings? <br> [3, 22] *And Lord said "Behold man has become like one of us, having the ability of knowing good and evil, and now, lest he stretch forth his hand and take also from the Tree of Life and eat and live forever".* <br> Has the independent living being got out of hand? Does s/he no longer pursue her/his mission? |
| Living matter transforms into nonliving matter through death and dissipation. <br> Living matter transforms into other living matter; it reproduces and mutates. <br> We are not currently able to transform nonliving, abiotoc matter into living matter, but only, for example, though incubation and artificial insemination. We did not eat from the tree of life. <br> The abilities to live and know are *separate*. |

### 3.1. Cohabitant Alienation

Alienation is generally assumed as a possible property for the spatially external, and is equivalent to strangeness or remoteness. Being alien has different meanings, such as coming from the outside, or being incommensurable, incompatible, incommunicable, and unrepresentable. What is alien is theoretically admissible and, in some way, convivable and even intersectional with the nonalien.

Alienation, culturally, often hides the presumption that there may be a master context and a sameness in all respects. This leads to geocentrism, anthropocentrism, the idea of dominant species, and the like.

Addiction to our way of life leads to seeing the enormous variety of other life as alien to us even if it is not alien to planet Earth; indeed, it is an integral part of it for evolutionary phases. We live with this alienity above all by using it, such as by eating it, making wood from it, and smelling its flowers. We may use living aliens such as when, for example, we make them work: for animal therapy; using their products, such as eggs, milk, and honey; setting up intensive animal husbandry; and testing drugs. In the same way, there is considered to be unconditional freedom in the use of nonliving things, although that freedom is today reduced by the phenomena of pollution and the exhaustion of resources.

We consider alien things inferior, not to mention nonliving matter, inferior. We would therefore have every right to use them (we are assumed to be in charge, see Table 1, with the conveniently reductive interpretation of the phrase *giving a name* misunderstood as an investiture, a delegation of use rather than the attribution of a responsible role) and to decide. After all, only humans, even if in different forms, have hope beyond death. Certainly not a dog, a bird, or a cat.

A theoretical and not just a benevolent concessive new philosophical reasoning is expected to allow for a conception of the world that is respectful of diversity and of life itself, with great self-ethical effects. This is the case with animal rights. How can a social thought be humanistic, self-caring, careful, values-based, and at the same time not conceive

of the rights of alien beings such as animals, and of the properties possessed or acquired by matter (see note 7)?

The lack of this attention leads to environmental problems, such as the reckless use of antibiotics in animals raised in unhygienic conditions, antibiotics that are then ingested by consumers; and the establishment of unknown and uncontrolled zoonotic diseases caused by pathogens jumping from animals to humans, such as the Ebola virus, salmonellosis, and COVID-19 (suspected to have been acquired from bats).

The anthropogenic destruction of ecosystems [35] to expand agriculture and human settlements is considered to reduce biodiversity, allowing more adaptable animals, such as bats and rats carrying zoonotic diseases, to proliferate. Often, it is a matter of the zoonosis of previously unrecognized diseases in populations lacking immunity.

Ethics is strategic and pays off in the long term, while unethical behavior can be catastrophic, often having irreversible effects, such as the use of resources, pollution, and exploitation.

*The thesis of this article is that the ethical, comprehensive understanding of the world is related, embedded, theoretically integrated in scientifically updated and humanistic interdisciplinary philosophical reasoning. We consider how alienation and impassability could facilitate the social adoption of such reasoning.*

Presumably, the peculiarities of our body, our context, its scale, and our habits would together form what we consider regularity.

Regularities also have different levels such as beauty and ugliness. We create beauty through music, paintings, and poems. Outside of recognized and created beauty, there would be deformations and monstrosities. Monstrosity and disgust are in turn assumed as signs of hostility and danger provided by Nature.

Furthermore, incommensurable, alien cognitive systems continue to cohabit with us. We may know a lot about bats, but we can never try to be a bat [12]. The same goes for all of the alien species we presumptuously live with.

Perhaps we will never be able to try to cognitively be the other; rather, they can only be represented in us.

We know that to be open we must start with being closed and always be able to re-close. Openness is not a matter of a lack of boundaries, but a choice.

Different scenarios may be simulated through virtual reality, cartoons, and video games, imitating our assumed understanding of the senses, vision, behavior, customs, and context of specific animals.

Music, literature, and art give us ways of thinking, words, and meanings with which to represent and design, freeing us from functionality and optimization only.

We are many, but substantially cognitively, existentially alone, in the sense that we can interact but maintain identity and specificity. Maintaining identity has the price of loneliness. Is the prospect of losing it with death a disaster or a liberation?

Identity [36] can be understood as a property emerging from our mind-body system (with variable bidirectionality), not only from its cognitive abilities that generate coherent self-recognition in consciousness and mind [37].

Identity, therefore, continues over time, not only due to the fluidity of its emergence, but also to possible irreducible multiplicity, such as in schizophrenia. It can be understood that the hypothesized process, the biological-cognitive converter (see Section 2), an ideal identity generator—presumably a system of processes of emergence—pauses during sleep and is extinguished with death, which therefore leads to the disappearance of this emerging identity [38,39]. This self-knowledge as a consciousness-generating mechanism is supposed to operate at various levels in the living, probably in correspondence with the levels of cognitive capacity [40].

In Genesis, we see the acquisition of identity, better understood as the acquisition of the property of generating identity, in Adam when he realizes that he is naked. He conceives of the environment in a different way; he realizes himself to be different and extends the ability to understand to a self-understanding. This is a matter of inappropriate

use: an improper extension of the understanding needed for survival, perceived as a *fault*. Adam hides himself; he does not rejoice in having achieved something that was not there before. He acts like a child in trouble after a prank. The situation is completely different from that of Prometheus, who steals fire from the gods to give it to human beings. The metaphor is of discovering, of understanding by stealing the secrets of Nature. "Stealing" because we did not have them before and because discovering and understanding are implicitly understood as unauthorized, with the connotation of violation (I am thinking of the first autopsies, the openings of sarcophagi and mummies, the first blood transfusion, and experiments asking Nature to phenomenologically respond). Adam, on the other hand, has nothing to give; he is ashamed.

The verb discover presupposes that something already existed, but was concealed from us. For instance, we believe that mathematics has always existed and was only waiting to be understood. This is different from considering mathematics as a field of flowers to be cared for, on which we might invent new grafts and cultures that were not there before. Objectivism and constructivism confront each other here. It is not a question of choosing, but of using both [6] (pp. 50–53) in a DYSAM-like way, by using the strategy of thinking based on how it is more convenient to think that something is rather than trying to find out how something really is [7] (pp. 76–79).

### 3.2. Describing Ourselves as Aliens

Trying to describe species' characteristics, particularly those of our species, with detachment, as if the subjects were alien, leads to surprising points of view. For instance, try describing walking, sleeping, eating, drinking, aging, crying, laughing, or sexual activity. It is difficult to try to explain such things to hypothetical aliens or in the design of simulations. One might also consider the stages of our life, from birth to death. This is the purview of medicine, which, however, soon discovers the limited effectiveness of objectivism alone, without the involvement of the "patient", who creates cognitive reality [40].

In the *Letter to Matter* above, I told her that "I know you want me this way" because my desires, my pleasures, and my actions are manifestations compatible with our degrees of freedom and with her evolutionary will, e.g., one sees oneself as sexually attractive when one is of reproductive age, one sees a newborn as a beautiful being that must be protected. It is amazing what Matter makes pleasurable at the right time. However, then nonfunctional beauty appeared—for example, music, painting, and art—do these have any evolutionary meaning?

The neurological [41], cognitive, and cultural bases of beauty must be explored without reductionism. Everything has to happen somehow, but what happens is often much more than *how* it happens.

For some reason, Nature has extended understanding from being necessary for life to allowing humans the ability to vary actions and decide, even if these are unfortunate decisions. Nature has played the card of understanding itself, but to what evolutionary advantage?

Are we conceivable as hypothetical Turing oracles in terms of biological algorithmicity, inserting an element of theoretical non-completability? In such a case, the Oracle should be understood to represent another interfering, incommensurable logic [42].

### 3.3. Impassability between Different Cognitive Systems

Experiences of levels of cognitive representations of impassability arise in limited intraspecies interactions, such as using behavioristic effects, training, taking advantage of natural behaviors (like beehives) and in unrealizable dreams where the admissibility criteria, for example of physics and logic, are diluted and even distorted, when the memory is *dreamt* and not just *thought* [43]. The insufficiency of evolution to explain the complexity of life (currently conjectured as originating from catalytic processes; see, for instance [44])

could be conceptually reduced when integrated with the phenomena of emergence [7] (pp. 255–258).

The impassability between cognitive systems is also to be considered with regard to the limits of thinkability [1]. For instance, asking ourselves about our existential meaning, which probably does not make sense, is not properly cognitively treatable.

In this regard, the intrinsic limitations of thinking and understanding can be considered in conjunction with theoretical incompleteness [2–4], the intrinsic non-completability of phenomena of various kinds, balancing *negotiations* between microscopic and mesoscopic[9].

Another aspect of impassability concerns the past, which can only partially be represented or remembered, and possibly understood but never relived, just as one cannot live as an alien species.

### 3.4. Physical and Existential Irreversibility

The experience of irreversibility as impassability is known and accepted. For example, in physics, some transformations are unidirectional, such as dissipation, diffusion, and dilution. Other examples are the wear of materials and physical-chemical transformations activated and supported by energy supplies, such as burning wood and paper, or cooking food.

This is thermodynamic unidirectionality, first investigated in the 1850s by Rudolf Clausius with the introduction of the concept of entropy.

Other cases involve, for example, the crushing of materials, which, if reconstituted, cannot regain their original properties, such as lenses and crystals.

Irreversibility usually relates to unrecoverable properties.

However, we live with generic and generalized irreversibility that is neglected as irrelevant at a macroscopic level, such as the molecules of poured and re-poured water, always different extinguished and rekindled fires, and living bodies that change continuously in a macroscopically irrelevant way on short timescales. We consider levels of representation that make irreversibility suitably negligible. Furthermore, irreversibility is used.

The irreversibility of the shattering of a lens and the cessation of the cognitive properties of the living after death have similarities and differences. They are both irrecoverable, though, in the first case, the properties were fixed, while in the second they were acquired continuously. The first case has major similarities with the irreversibility of dissipation and dilution. Experience after death, if any, should not be detected by our identity, which is lost with the end of the process of emergence. How then could after-death experience be detected? Does it exist? [19,34]

With reference to acquired properties, for example, emerging ones, irreversibility usually refers to cases of their nonrecoverable loss, as when the process of emergence, for whatever reason, ceases to be active (emergent properties do not result, but are continuously generated). However, the processes of emergence generate uniqueness such as identities [36]; parts of the body can have complete molecular replacements but still be recognized as the same; collective phenomena are recognized as having stable identity even if structurally they are continuously different, such as a flock being recognized as such even if it continuously acquires different shapes, and a cytoskeleton being characterized by structural dynamics since its parts are continuously destroyed, renewed, or newly created.

Reversibility is, in reality, almost conceptual, hypothetical, and partial while, when occurring, irreversibility is almost complete, unrecoverable, with the two properties being dependent on the level of description.

Death is conceptually related to the common fact of cessation of specific processes of emergence, such as the functioning of devices, the exhaustion of resources, and the life cycles of fundamental species, e.g., in agriculture. The death of individuals is considered

---

[9]　The microscopic level is intended to refer to minimal, indivisible constituents. The macroscopic level is intended to refer to functional aggregations in which the microscopic is no longer considered. The mesoscopic level is intended to refer to aggregations of various microscopic levels in particular clusters; see [7], pp. 110–116.

implicitly painful but acceptable, unavoidable, and indeed necessary to make room for new births. However, the death of a species either as extinction or through mutations is often considered avoidable, but still accepted as a phenomenological need.

The irreversibility of death for living beings occurs in various ways such as the widespread and gradual cessation of functionality and, in some cases, the sudden cessation of cognitive functions.

We notice how the cessation of cognitive properties can be detected though the cognitive properties of observers. For all we know, death is not self-detectable.

Is it meaningful to consider the problem of its comprehensibility? Is it like trying to measure something with an inadequate, incommensurable unit of measurement like surfaces in liters? These are problems like understanding the understanding [45].

We are uncomfortable with the irreversibility of death for human beings and loved animals.

For quite some time now, the euphemism "to pass away" has been in use instead of "to die". Is this a way to differentiate the biological death of humans from that of animals?

People stopped "dying" many years ago. Instead of dying, they "pass away" or "pass on" (probably in conceptual reference to the passage through the Red Sea in Jewish culture). Today, they merely "pass." Does this euphemism implicitly contain transitional aspects to be understood as hope? Is death understandable as the occurring of a phase transition-like process [46], as considered below?

### 3.5. The Impassability and Irreversibility of Life and Death

The theme of impassability and irreversibility concerns issues such as the Big Bang—asking about what came before makes no sense. However, there may be properties that can be considered conceptually as residual [47], such as the intelligence [48] of matter, e.g., the ability to condense and self-organize; the origin of life, tentatively considered today as based on autocatalysis (molecules that catalyze their own or reciprocal replication, in a mutually advantageous way; see [44]); and the ability to evolve [49].

In particular, this impassability concerns the meaning of death (or, rather, the completion of the process of dying) and what eventually comes after [33]. Currently, the after is usually considered in an elementary, pre-relativistic, and prequantum way. However, let us consider here its semantic, existential meaning. I wonder about this at a mature age, with the depth that Matter-Nature allows and perhaps wants. The time is ripe to start thinking about death: memories are now greater than projects.

Beyond religious speculations, which have their importance, I note that the notion of after death as alienation reveals aspects of incommensurability, probable unthinkability, and inconceivability, in some way related to the alienation of other forms of life, especially their cognitive systems and experienced normality, as discussed above. We face the oxymoron of comprehending that something is theoretically incomprehensible, while incomprehensibility only refers to the severe limitations of our knowledge.

Customs say that certain acquired relationships, such as marriage, are valid until death, but not natural ties such as being a mother, father, child, brother, or sister.

Hypothetically, granted the persistence of something after death, this would have to be other than the role played by biological life and there would be a general cessation of identity [36]. We have always thought of ourselves as humans (but starting from what evolutionary stage?).

In essence, the cessation of identity, with death conceived of as a painful, inevitable end, would instead be a point of arrival or even reward, as assumed in some cultures and religions, where existence after death is bestowed according to credits. Such phase transitions (or metastability) are supposed to occur given adequate contextual physical situations and adequate contextual, energy levels[10] [46].

---

[10]    From Matthew 13:9: "Whoever has will be given more, and he will have an abundance. Whoever does not have, even what he has will be taken from him".

## 4. Conclusions

The incomprehensibility of death is not unique, but comparable with other situations. Experiences of impassability would be like *dying a little*, as they would introduce us to the humiliation of not understanding.

We should, however, note that the philosophical reasoning outlined above is still based on concepts, meanings, and representations that may be scientifically misaligned or outdated (for example, the planetary, so-called plum pudding model of the atom and the Christian beyond in the collective imaginary are still substantially similar to what was outlined by Dante Alighieri in his *Divine Comedy*) in reference to the principles and concepts currently used—for example, in physics, considering fields rather than objects [50] and quantum understandings [25,51]. The social imaginary is still based on the Newtonian vision [52] of the separability, stability, replicability, distinguishability, and adjustability of objects and forces. This vision has been consolidated and confirmed in literary, poetic, artistic, and philosophical representations. Artistic and philosophical representations, as well as interpretations related to new concepts and their generative mechanisms, are necessary to allow for more adequate ways of reasoning and representing, and for better philosophical reasoning. Such a new vision and the related philosophical reasoning is expected to allow higher levels of awareness and lead to more comprehensive conceptions of the world that are no longer utilitarian, but respectful of diversity and having positive ethical effects.

This is the case for plants and animals, perceived as aliens but still having rights, and for ethics in general, which should be strategic and convenient in the long term, while unethical actions can be catastrophic, as in the cases of unsustainability and exploitation.

The thesis of this article was that the ethical, comprehensive understanding of the world is related, embedded, and theoretically integrated in scientifically updated and humanistic interdisciplinary metalevel philosophical reasoning [19] (p. 63). We consider how alienation and impassability could facilitate the social adoption of such reasoning.

We end by paraphrasing Von Foerster, who said that there are no anomalies in the environment. If a given phenomenon looks strange and incomprehensible, it means that the theoretical framework used is inadequate. This requires us to create new conceptual frameworks [46]. In this regard, we propose the adoption of new philosophical understandings, updated to be interdisciplinary, with the science of complexity and humanistic views, together allowing for the emergence of a social imaginary and having ethical effectiveness.

**Funding:** This research received no external funding.

**Conflicts of Interest:** The author declares no conflict of interest.

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
