# Peer review of "Letter to Matter and Various Incomprehensibilities—The Effective Ethicality of Scientific and Humanistic Interdisciplinarityâ€"

_philosophies, doi:10.3390/philosophies6020026_

Round 1
Reviewer 1 Report
This is an important contribution to thought from someone with a mastery of systems science and philosophy. It is a "letter" written by a human to other humans, who die as well as live. The author brings alive the historically important concept of the clinamen, also referred to by Igamberdiev, as well as many others.
I have just three suggestions which the author should have no difficulty in implementing:
1) The Title is a bit strange; better would be simply Letter ON Matter. The usage "on" is a familiar one from the time when such letters were common. "Matter" is not capable of actually receiving our messages!
2) "Interdisciplinarity" is a term that suffers from lack of context: why "inter" rather than "multi-" or "trans-"? The view of Nicolescu is that all three are "arrows in the quiver of knowledge". In my view thie author goes beyond the simple juxtaposition of disciplines or their joint use to something much more basic. This is important for the acceptance of the valuable thesis of the paper. A one sentence definition would be fine.
3) Quasi-ness is a term that is familiar to me but perhaps less so to others. My comment here is the same a for 2) - just a little additional treatment to position quasi-ness as a (necessary) argument against classical binary descriptions of reality.
Author Response
Reviewer 1
Dear Reviewer,
Thank you for your comments very appreciated.
- The Title is a bit strange; better would be simply Letter ON Matter. The usage "on" is a familiar one from the time when such letters were common. "Matter" is not capable of actually receiving our messages!
I would like to keep the version “Letter to Matter” because in the text Matter is metaphorically personalized such as when I write “I know you want me that way…“, page 6 Section 2.
- "Interdisciplinarity" is a term that suffers from lack of context: why "inter" rather than "multi-" or "trans-"? The view of Nicolescu is that all three are "arrows in the quiver of knowledge". In my view thie author goes beyond the simple juxtaposition of disciplines or their joint use to something much more basic. This is important for the acceptance of the valuable thesis of the paper. A one sentence definition would be fine.
The subject of interdisciplinarity is elaborated at page 3. I added text specifying why not multi- or trans-.
- Quasi-ness is a term that is familiar to me but perhaps less so to others. My comment here is the same a for 2) - just a little additional treatment to position quasi-ness as a (necessary) argument against classical binary descriptions of reality.
The subject is introduced at page 2. I inserted new explicative text.
Sincerely
The author
Reviewer 2 Report
Thank you for this interesting reading. The relationship between "theoretical incompleteness in systems science and theoretical incomprehensibility" is full of implications and it is well presented and discussed. Several stimulus for reflections can be derived from this paper.
Author Response
Reviewer 2
Dear Reviewer,
Thank you for your evaluation; it is most appreciated.
Sincerely,
The author
Reviewer 3 Report
The Author is a systemic scientist and a mathematician. He feels the need to open his research to philosophy, with an interdisciplinary and cooperative approach aimed at dealing with existential problems, traditionally falling within philosophical competence, such as: matter, death, ethics, god, reasoning, comprehension.
By means of an interdisciplinary approach, the Author makes some suggestions to philosophers, while he acquires some issues from philosophy.
He suggests that philosophy should abandon a vision of the world based on Newtonian physics, as well as update its references to contemporary sciences. This involves giving up many concepts, such as completeness, logical closure, and acquiring new ones, such as complexity, systems, incompleteness, logical openness.
He acquires from philosophy the sensibility towards existential problems to be faced with philosophical expertise.
I very much appreciate the Author’s proposal and I definitely agree with the demand for a reciprocal reference update.
Given that the focal point of the essay consists in a proposal for an interdisciplinary research, where natural and human sciences crossbreed their knowledge and mutually update their references, and given that a scientist cannot become an expert in philosophy – just as a philosopher will not manage to fully possess specialistic knowledge of any science – it becomes indispensable that some kind of direct cooperation among scientists and philosophers finds its way in research, being subsequently implemented in literature.
How can such cooperation be realized? Following the path opened by the Author, I imagine philosophers and scientists seating at the same table confronting their ideas over a common subject (“matter” could be a good topic for an interdisciplinary session). Scientists will contribute handing over to philosophers updated scientific discussion and/or discoveries and philosophers will contribute making explicit pieces of philosophy often embedded in scientific research, and also following the ethical, or social, or even political consequences of research programs.
In this collaborative attitude I would like comment from a philosophical point of view on the following three questions addressed by the Author:
- The Author refers to “philosophical reasoning” and this expression could suggest that philosophy makes use of a special and proper form of reasoning. I would like to stress that “reasoning” doesn’t acquire particular features in philosophy, nor in other domains: “reasoning” is a universal conceptual tool used by humans in any domain to understand and explain problematic features of the world. There is not such a thing as “philosophical reasoning”: there is only “reasoning”. When speaking of philosophical, or physical, or legal, or any other form of “reasoning”, the difference is given by domain of exercise, not by the conceptual tool being used.
- Comprehending that something is incomprehensible is an oxymoron, says the Author. This is true if we claim that to say that something is incomprehensible – let’s think of death, or of coronavirus – we need to largely understand it. But this is not always the case: we can understand that something is incomprehensible because we observe a phenomenon and we are able to name it, or identify it, but cannot explicate nor describe it. Incomprehensibility only refers to the severe limitations of our knowledge. We can comprehend that something is incomprehensible, without any contradiction.
- In Letter to Matter, the Author seems to be longing to escape the boundaries of our subjective point of observation. Such escape is an impossible task, because whatever is said, observed, referred, spoken of, thought, is said, observed, referred, spoken of, thought, by a subject, with his/her subjective powers and limits. Being subjective doesn’t imply being egoistic or egocentric: I cannot know how a bat feels, but this doesn’t imply that I can kill it, use it for vivisection experiments or try to destroy the species “bat”. Feeling to be myself is not negationable: it is a fundamental feature of our constitution that is to be taken as it is: no room for complaint, nor for imagining revisionary ontologies that would only exhibit the width of our imagination, without adding nothing to our comprehension of what there is.

Author Response
Reviewer 3
Dear Reviewer,
Thank you very much for your helpful comments and suggestions.
In this collaborative attitude I would like comment from a philosophical point of view on the following three questions addressed by the Author:
In reference to point 1 I added text at page 3 and page 15 on philosophical reasoning intended as suitable meta-level
In reference to point 2 I added text at page 14
In reference to point 3 I added text at page 3.
Again, many thanks for your suggestions.
Sincerely,
The author